# Development of Gelatin/Misoprostol Compounds for Use in Pregnancy Failures

**DOI:** 10.3390/ma14237250

**Published:** 2021-11-27

**Authors:** Thiago Cajú Pedrosa, Rossanna Trócolli, Wladymyr Jefferson Bacalhau de Sousa, Glauber Rodrigues Cerqueira de Cerqueira, Henrique Nunes da Silva, Rossemberg Cardoso Barbosa, Matheus Ferreira de Souza, Taynah Pereira Galdino, Jackeline Nascimento Apolori Tissiani, Marcus Vinícius Lia Fook

**Affiliations:** Laboratório de Avaliação e Desenvolvimento de Biomateriais do Nordeste–CERTBIO/UFCG, Campina Grande 58429-900, PB, Brazil; thiago.cajupedrosa@gmail.com (T.C.P.); rossannatrocolli@gmail.com (R.T.); wladymyrjb@yahoo.com.br (W.J.B.d.S.); henrique.nunes@certbio.ufcg.edu.br (H.N.d.S.); rcbvet@gmail.com (R.C.B.); fdsmatheus@gmail.com (M.F.d.S.); taynahgaldino01@gmail.com (T.P.G.); apolori@hotmail.com (J.N.A.T.); marcus.liafook@certbio.ufcg.edu.br (M.V.L.F.)

**Keywords:** gelatin, misoprostol, sodium tripolyphosphate, lyophilization, abortion, uterine emptying, biocompatibility

## Abstract

Early abortion is one of the most common complications during pregnancy. However, the frequent handling of the genital region, more precisely the vagina, which causes discomfort to patients in this abortion process due to the frequency of drug insertion, as four pills are inserted every six hours, has led to the search for alternatives to alleviate the suffering caused by this practice in patients who are already in a shaken emotional state. Hence, this work aimed to develop composites of gelatin and misoprostol, using a conventional single-dose drug delivery system. These composites were prepared by freeze/lyophilization technique, by dissolving the gelatin in distilled water, with a concentration of 2.5% (*w*/*v*), and misoprostol was incorporated into the gelatin solution at the therapeutic concentration (800 mcg). They were subsequently molded, frozen and lyophilized. The samples of the composites were then crosslinked with sodium tripolyphosphate (TPP) 1% (*v*/*v*) with respect to the gelatin mass for 5 min. The characterization techniques used were: Optical Microscopy (OM), Fourier Transformed Infrared Spectroscopy (FTIR), Thermogravimetry (TG), Swelling, Biodegradation and Cytotoxicity. In OM it was observed that the addition of the drug improved the cylindrical appearance of the compounds, in comparison with the sample that was composed of only gelatin. There was a reduction in the degree of swelling with the addition of the drug and crosslinking. The cytotoxicity test indicated the biocompatibility of the material. Based on the results obtained in these tests, the composites have therapeutic potential for uterine emptying in pregnancy failures, especially in the first trimester.

## 1. Introduction

Throughout the twentieth century, supported by the discovery of the contraceptive pill and by the separation between reproduction and pleasure, women’s right to reproductive and sexual health gained space, and several changes were observed due to feminist movements and claims by social groups [1]. Approximately 40% of all pregnancies are not intended or planned, leaving one to face unwanted birth or terminate the pregnancy through abortion [2].

Abortion in Brazil is legal when pregnancy is as a result of rape, when there is risk to the pregnant woman´s life (therapeutic abortion), or when the fetus has a malformation incompatible with life, such as anencephaly. The illegal practice of abortion is one of the factors that makes this procedure a serious public health problem [1,2]. The drugs that are mostly used with safety to remedy cases of pregnancy termination are RU 486 and misoprostol.

Misoprostol is a synthetic analogue of prostaglandin E1 (EP1), and it is administered vaginally by the manual insertion of tablets [3]. The dosage commonly used for uterine emptying is four 200 µg tablets, a total of 800 µg, repeated at an interval of four to six hours [4]. However, daily practice shows that the fact that the protocol indicates the use of 800 µg of this drug in pregnancy failures, and that current presentations do not exceed 200 µg, causes situations that present greater discomfort to patients who need this drug. This discomfort is often related to edema of the vulva, local pain, erythema, asthenia, and anxiety caused by both excessive handling and length of stay. To facilitate administration and offer more comfort to women, misoprostol can be added to biodegradable and biocompatible components, such as gelatin, to obtain a biocomposite.

Gelatin is a purified protein obtained by partial acid hydrolysis (Type A) or partial alkaline hydrolysis (Type B) of animal collagen and can consist of a mixture of the two types. Gelatin has aroused great interest for the production of composites and the transport of drugs; this is justified by its being an abundant raw material, produced practically all over the world at a relatively low cost, and for having excellent functional properties and the ability to form fibers [5,6].

Arcanjo et al. [7] evaluated the efficiency of administering misoprostol intravaginally to promote uterine emptying in early interrupted pregnancies, certifying that its vaginal use is a safe and effective alternative to uterine curettage for interrupted pregnancies. Parmar et al. [8] conducted a comparative study using 25 µg of misoprostol vaginally with intracervical cerviprime gel to assess the effectiveness of the drug’s use, confirming that the use of misoprostol lad how induction for the delivery interval, reducing maternal complications. Huang et al. [9] evaluated the effects of sequential vaginal and sublingual misoprostol for the treatment of abortion in the second trimester of pregnancy, finding an adequate alternative for use in abortion procedures.

The reported studies with misoprostol were conducted with the conventional dosage; furthermore, there have been no studies on reducing the vaginal handling imposed on patients, exposing them to infections and discomfort caused by the usual procedure. Therefore, the objective of this work was to obtain a biocomposite of gelatin and misoprostol (800 µg), for administration through a conventional single-dose drug delivery system, to minimize the pain caused to patients who need uterine emptying procedures.

## 2. Materials and Methods

### 2.1. Materials

Type A gelatin (obtained from porcine skin), misoprostol and sodium tripolyphosphate (TPP). Sigma Aldrich ^®^ (Suzano, SP, Brazil).Citric Acid. Vetec (São Paulo, SP, Brazil)0.9% saline. Adv Farma (Nova Odessa, SP, Brazil).

### 2.2. Methods

#### 2.2.1. Preparation of Gelatin Egg Cells

The gelatin egg cells biomaterial was prepared using a freeze-drying technique. The 2.5% *w*/*v* solution was prepared by dissolving the polymer in distilled water at 40 °C under mechanical stirring (160 rpm) for 1 h. The solution was frozen at −68 °C for 96 h and lyophilized for 48 h. Then, it was immersed for 5 min in solutions of sodium tripolyphosphate (TPP) at 1% (*w*/*w*, based on the mass of the samples), frozen (−68 °C, 96 h) and lyophilized (48 h) to obtain the crosslinked product.

#### 2.2.2. Preparation of Gelatin/Misoprostol Egg Cells Compounds

After obtaining the solubilized gelatin egg cell, the drug (misoprostol 800 mcg/mL) was added and homogenized for 2 min at 40 °C under mechanical addition (160 rpm). The solution containing it was frozen at −68 °C for 96 h and lyophilized for 48 h to obtain the gelatin/misoprostol composite. Then the biomaterial was crosslinked with sodium tripolyphosphate (TPP) following the same methodology used in gelatin without drug. The gelatin/misoprostol compound after crosslinking was again swollen with distilled water, frozen (−68 °C, 96 h) and lyophilized (48 h) to maintain its shape.

### 2.3. Characterization

#### 2.3.1. Optical Microscopy (OM)

Optical Microscopy was used to evaluate the morphology and microscopic aspects of the composites obtained. The equipment used was a Hirox Optial Microscope (Hackensack, NJ, USA) for reflection and transmission with 2D accessories and magnification variation from 20× to 2000×, coupled to an image analysis station.

#### 2.3.2. Fourier Transformed Infrared Spectroscopy (FTIR)

The samples were submitted to Fourier Transform Infrared Spectroscopy (FTIR, Waltham, MA, USA), at room temperature (24 °C) and the equipment used was Perkin Elmer´s Spectrum 400 (Waltham, MA, USA). The FTIR technique was performed to identify the bands, characteristics of the adaptable groups, present and possible interaction of the drug with the polymeric matrix, using a measurement range from 4000 to 650 cm^−1^.

#### 2.3.3. Thermogravimetric Analysis (TGA)

The thermogravimetric analysis (TGA) of the samples was performed in a Perkin Elmer model Pyris 1 TGA equipment (Shelton, Washington, DC, USA), with a material quantity of approximately 5 mg; the sample were weighed on a precision scale (±0.1 mg). The material was heated with a heating rate of 10 °C/min, under a nitrogen atmosphere with a flow rate of 50 mL/min, in an aluminum crucible.

#### 2.3.4. Swelling Test

This test was carried out with the objective of evaluating the degree of swelling of the obtained sample, which was initially weighed and kept in saline solution for 30 min. Then, the samples were removed from the solution, placed on a filter paper to remove excess solution and weighed on a digital scale. All samples were subjected to measurements before and after swelling. Twenty samples were used, 10 with the drug. The degree of swelling of each sample at time t was calculated using Equation (1):(1)$$GI\left(\%\right)\;=\;\frac{W_{T}-W_{O}}{W_{O}}\;\times 100$$
where *W_T_* is the weight of the sample at time *t* and *W_O_* is the start weight of the sample.

#### 2.3.5. Biodegradation

The biodegradation test was performed to observe the behavior of the sample when immersed in a 0.0055 M citric acid solution with pH 4.0 and 37 °C, simulating the intravaginal environment. The result was obtained from the mass ratio using Equation (2):(2)$$MR\left(\%\right)\;=\;\frac{M_{i}-M_{f}}{M_{i}}\;\times 100$$
where *M_i_* is the start weight of sample and *M_f_* is the final weight of sample.

#### 2.3.6. Cytotoxicity

The in vitro cytotoxicity evaluation of the samples was carried out by the MTT cell viability assessment test [3-(4,5-dimethylthiazol-2-yl)] according to ISO 10993-5:2009 [10]. The cell line was L929, acquired from the Cell Bank of the Federal University of Rio de Janeiro (Rio de Janeiro, Brazil) and the evaluation parameters observed are the percentage of cell death and the IC50 (concentration of the product that inhibits 50% of growth cell). The analyses were performed at the Laboratory for Evaluation and Development of Biomaterials of the Northeast–CERTBIO (Campina Grande, Brazil), laboratory accredited by ABNT ISO/IEC 17025:2005, CRL 0799 for Chemical and Biological Tests.

## 3. Results and Discussion

### 3.1. Optical Microscopy (OM)

Figure 1 illustrates the results of optical microscopy of gelatin samples with and without the addition of the drug misoprostol and crosslinked or not with TPP. The addition of the drug influenced the conformation of the material, as a uniform surface and more cylindrical shape are observed, indicating interaction between the constituents. It also verified that the crosslinking process with TPP caused the sample to retract after the second lyophilization cycle. This effect is due to the closer approximation of the polymer chain with the crosslinking process, making the structure of the compound more compact. The loss of regularity in geometry of the crosslinked compounds can also be related to the partial dissolution of gelatin during the crosslinking process.

### 3.2. Fourier Transformed Infrared Spectroscopy (FTIR)

Figure 2 illustrates the characteristic bands of powdered gelatin. In the bands at 3300 cm^−1^, 3080 cm^−1^ and 2940 cm^−1^, axial deformations of the OH stretch, of the associated NH in amides and of aliphatic C–H, respectively, were observed. At 1630 cm^−1^ and 1540 cm^−1^ angular deformations of C=O of N-substituted amides and symmetrical angular deformation of NH_2_. It was possible to notice angular deformation of CH_2_ at 1450 cm^−1^ and in the region 1390 cm^−1^ vibrations of the angular deformation type of CH_3_. In the band at 1330 cm^−1^ angular vibrations are observed in the plane of the OH bond, in 1240 cm^−1^ and 1080 cm^−1^ absorption of unconjugated C–N bond, in primary and secondary amines [11].

Figure 3 illustrates the characteristic bands of misoprostol. In the bands at 3500–3200 cm^−1^, there is an axial deformation of the associates O–H. In the region of 2940–2850 cm^−1^, axial deformation of the aliphatic C–H bond occurs. At 1658 cm^−1^ it is attributed to axial deformation vibrations of C=C double bonds. In the band referring to the region 1428 cm^−1^ there is an angular deformation of CH_2_ adjacent to carbonyl and in 1386 cm^−1^ angular deformation of CH_3_. In the region 1330–1260 cm^−1^, angular vibration in the connection plane and axial stretching vibrations of the OH connection are observed. In the region at 1180–1030 cm^−1^, there are angular C–O deformations of ethers. At 990–913 cm^−1^ angular deformation is noted outside the plane of the RCH=CH_2_ group and, at 880 cm^−1^, vibrations of the C–H bond outside the plane of the R2C=CHR group are observed [12].

In Figure 4 the FTIR spectra of the gelatin/misoprostol and gelatin/misoprostol/TPP compounds are shown. The presence of characteristic peaks of the drug in 2930 cm^−1^, 1381 cm^−1^ and 1050 cm^−1^ is observed, referring to the folding and elongation of the groups present in the molecule [13], as observed in the FTIR result of misoprostol, where the chemical nature of the drug used in the research was proven. Characteristic bands of gelatin are also observed, but at lower intensities. Such results confirm the incorporation of the drug in the gelatin matrix.

### 3.3. Thermogravimetric Analysis (TGA)

Figure 5 illustrates a comparison between the TGA curves in relation to the material obtained with and without drug, crosslinked and not with TPP. An initial loss of mass was observed between 50 and 90 °C, mainly due to the loss of water molecules. The second loss was between 250 and 300 °C and was due to the destabilization of the macromolecule, leading to thermal degradation. Liu et al. [14] analyzed the TGA of gelatin and its result showed a mass loss event close to 100 °C, probably related to the loss of water molecules present in the sample, and the second stage was related to degradation that occurred close to 330 °C. Data similar to the work reported by Mishra et al. [15]. 

The material with and without crosslinking exhibited a similar pyrolytic pattern, with small differences in the degradation profile. It is observed that the crosslinked samples showed grater thermal stability, which proves their crosslinking [16]. The sample with drug and without crosslinking was similar to samples without drug addition, with only one shift at the beginning of mass loss events to lower temperatures. The gelatin/misoprostol sample crosslinked with TPP showed loss of mass when compared to the others.

### 3.4. Swelling Test

The degree of swelling of a material can be defined as the amount of solvent absorbed by it. This property is related to the structure of the network formed in the crosslinks in the polymerization and crosslinking process of the polymer, observed through the crosslinking density [17]. The expansion capacity of biomaterials is an important aspect to evaluate their properties when applied as controlled drug delivery systems [18]. Figure 6 illustrates the degree of swelling of the gelatin samples with and without misoprostol, crosslinked and non-crosslinked.

It was observed that the biomaterial submitted to the crosslinking process showed a higher density and consequently a decrease in swelling, this behavior was also observed in the literature, where it is reported that the swelling of gelatin decreases with the crosslinking process [19]. Sudrajat et al. [20] found similar results. In addition, the incorporation of the drug decreased the ability of the gelatin matrix to absorb the PBS solution, which may suggest a subtle decrease in the hydrophilic character of the gelatin or intermolecular interactions between the drug molecules and the gelatin chains. According to Bona [21], the swelling index is directly related to the solubility in water, being an important parameter for the knowledge of the general characteristics of a material, mainly in relation to its resistance in aqueous medium, which can be confirmed by him when he affirms that solubility is determined by the chemical structure of the material.

### 3.5. Biodegradation

The enzymatic hydrolysis of a polymer will depend on the hydrophilicity of the polymer chain. This hydrolysis should always occur on the surface of the polymer and can be facilitated by increasing its surface hydrophilicity [22]. Biodegradation is an important property in the development of drug delivery systems, since it is one of the mechanisms that control the release kinetics [23]. According to Dallan [24], the evaluation of the degradation of a biomaterial in the biological environment is of paramount importance, since this characterization is directly related to the time in which the material will be absorbed by the organism after its insertion in the human or animal body. In this sense, the degradation test was carried out to investigate the stability of the composites and their loss of mass over time. Table 1 illustrates the results of the biodegradation test.

The composites without crosslinking solubilized more than 85% in the period of up to 30 min, whereas the sample of pure gelatin was completely solubilized in that same period.

TPP has the crosslinking effect, as it holds the polymeric chains, thus causing less water permeability and consequently less swelling and biodegradation. The large number of negative charges of TPP results in its ability to ionically crosslink some polymer such as gelatin [25]. The process of crosslinking and adding the drug provided a slight stability of the system in an acid medium. The study is promising because these compounds should be solubilized as soon as possible, promoting the release of the drug and, consequently, causing the dilation of the cervix. This behavior corroborates the results reported by Shankar et al. [26].

### 3.6. Cytotoxicity

The results of the cytotoxicity test are illustrated in Figure 7, where it is possible to observe that the material studied has biocompatibility. The adhesion caused between the gelatin and gelatin/misoprostol/TPP scaffolds and the cells comes from the linkages of the integrins present in the cells across the cytoplasmic membrane. This behavior in the structures is correlated in gelatin in studies. It is seen that in all cases the viability of L929 fibroblasts was above 70%, as required by the ISO standard [10] to consider non-toxic forine virotests. Furthermore, the increase in enzyme activity during the preparation of the tests did not cause a significant decrease in cell viability [27]. From the results, it can be said that the incorporation of the drug and the crosslinking of the matrix do not affect the biocompatibility of gelatin. Chang et al. [28] found similar results.

## 4. Conclusions

The results of OM indicate that addition of misoprostol positively influenced the shape and appearance of the biomaterial; in addition, the crosslinking process with TPP promoted a retraction in the samples after the second lyophilization cycle. FTIR spectra of the material with and without crosslinking with TPP showed the incorporation of the drug in the gelatin matrix. Thermogravimetric curves reveal that the compounds crosslinking with TPP have slight greater thermal stability. Regarding the biocompatibility properties, the process of crosslinking and adding the drug increased the solubilization time of the biomaterial, even so, it still remained with total solubilization in acid medium in a maximum 1 h. As for the cytotoxicity test, the biomaterial showed cell viability within the current standards. Therefore, it can be concluded that the compounds have potential for use in the release of misoprostol in order to reduce the vaginal handling imposed on patients, exposing them to infections and discomfort caused by usual procedure, the partition or loss of the recommended drug, thus minimizing both the risks of therapeutic failure and the length of hospital stay and aspects related to dosing or insertion errors.

## Figures and Tables

**Figure 1 materials-14-07250-f001:**
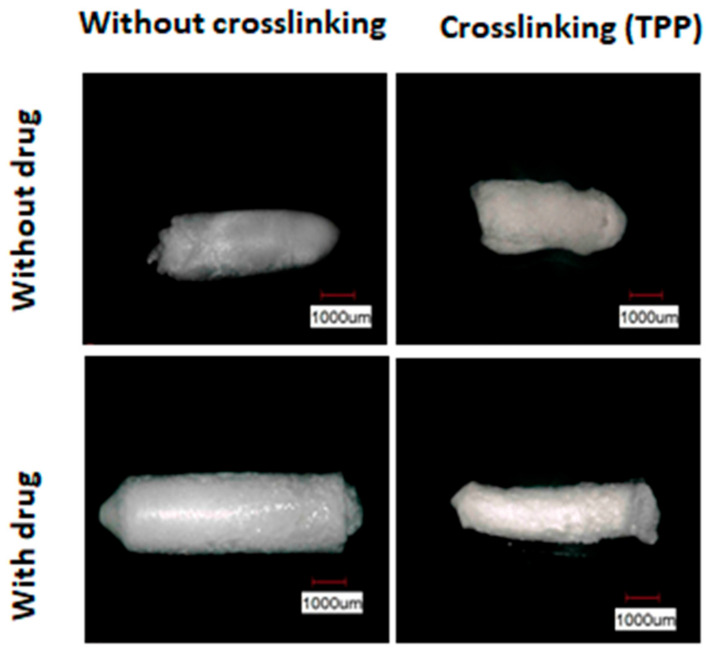
OM images obtained for gelatin/misoprostol compounds with and without crosslinking with TPP. The images are at the same magnification and the magnification bar (blank) is equivalent to 1 mm.

**Figure 2 materials-14-07250-f002:**
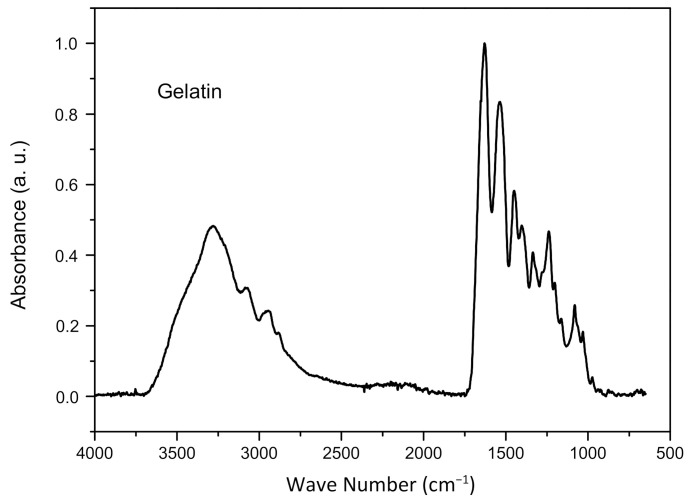
FTIR of gelatin.

**Figure 3 materials-14-07250-f003:**
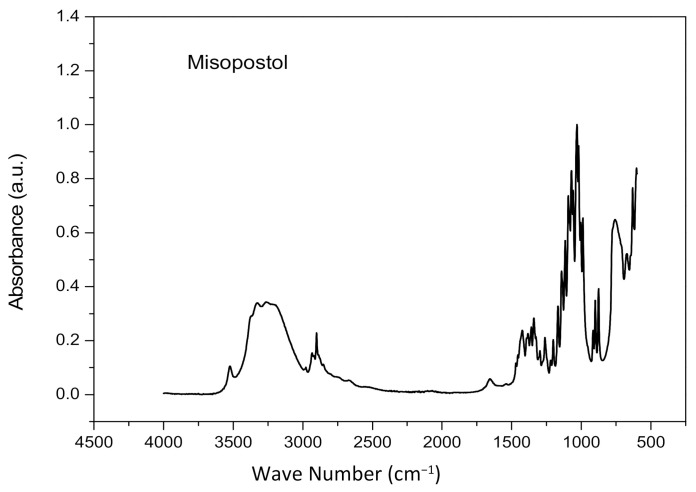
FTIR of misoprostol.

**Figure 4 materials-14-07250-f004:**
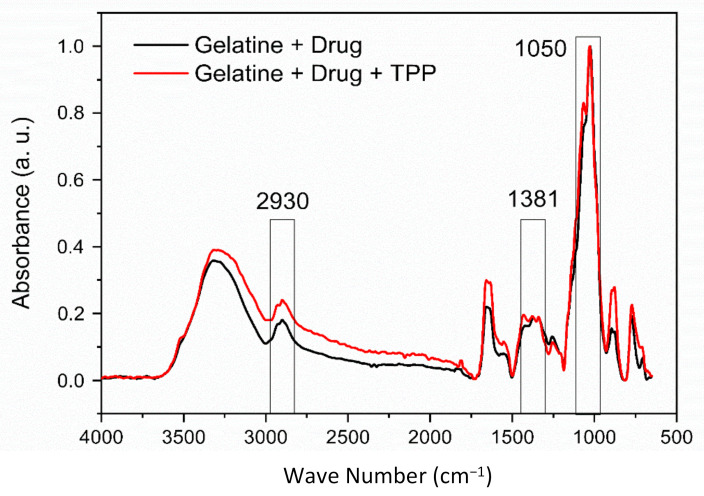
FTIR spectra of gelatin/misoprostol compounds with and without crosslinking with TPP.

**Figure 5 materials-14-07250-f005:**
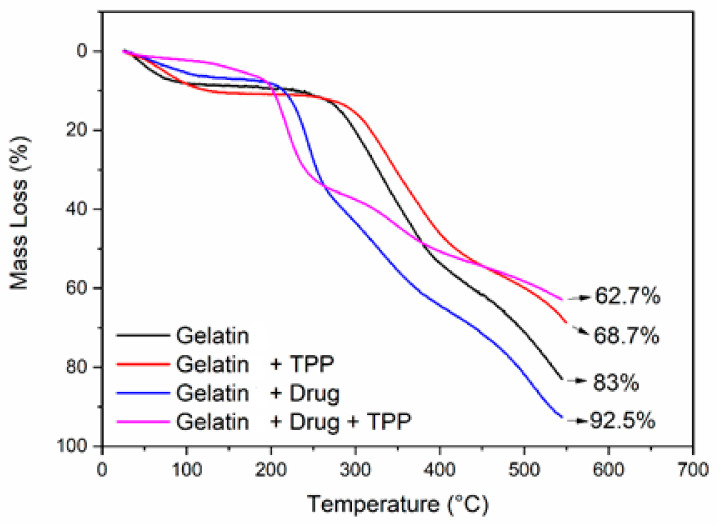
TGA curves obtained for gelatin/misoprostol compounds with and without crosslinking with TPP.

**Figure 6 materials-14-07250-f006:**
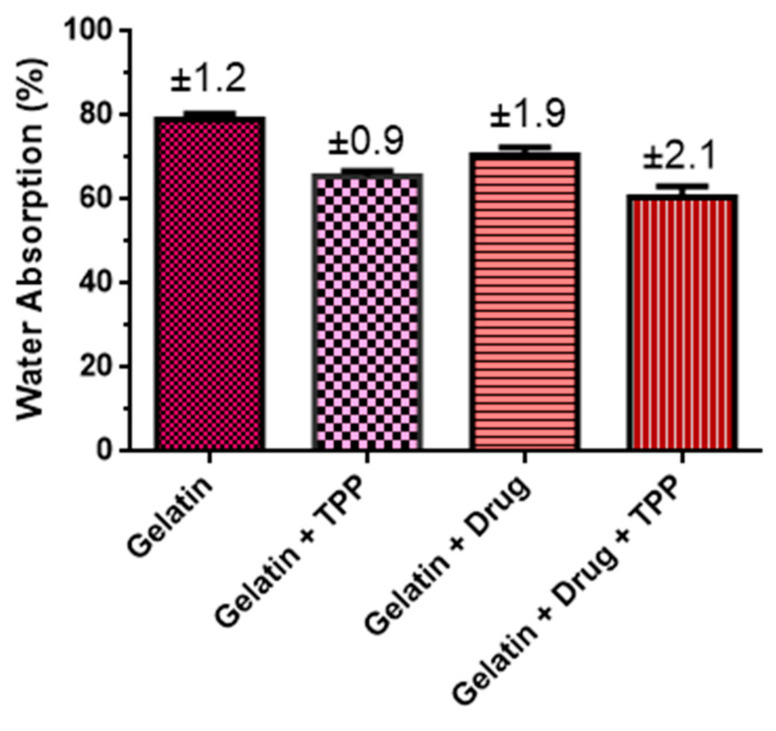
Degree of swelling of the compounds.

**Figure 7 materials-14-07250-f007:**
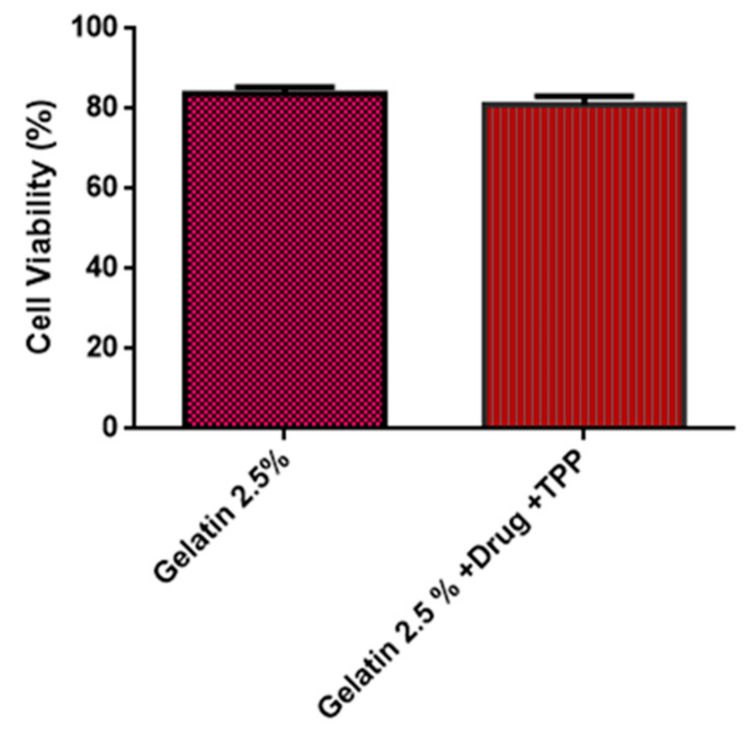
Cell viability of compounds.

**Table 1 materials-14-07250-t001:** Biodegradation in acidic solution of the compounds within a maximum interval of two hours.

Samples	Loss of Mass of Compounds (%)	
	Time	
	30 min	1 h
Gelatin 2.5%	100%	-
Gelatin 2.5% + TPP	87% ± 2.7	100%
Gelatin 2.5% + Drug	91% ± 4.3	100%
Gelatin 2.5% + Drug + TPP	86% ± 2.9	100%

## Data Availability

The data presented in this study are available on request from the corresponding author.

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
