# Peer review of "Development of Gelatin/Misoprostol Compounds for Use in Pregnancy Failures"

_materials, 2021, doi:10.3390/ma14237250_

Round 1

Reviewer 1 Report

I recommend the paper for publication. Nevertheless, some improvements should be made:

  1. Key words: instead of “TPP” “sodium tripolyphosphate” is better to use. Further, some other key words should be added, such as “biocompatibility”, “uterine emptying” or “matrice”. What do you think?
  2. Page 2, line 80: Before “Type A…” delete a full stop.
  3. Page 4, line 160, Figure 1: The abbreviation for optical microscopy, as previously described, is “OM”, not “MO”.
  4. Page 6, discussion referring to FTIR spectra of compounds: In the text the peak number 2940 cm-1 does not correspond to the peak in Figure 4, number 2930 cm-1. Correct it.
  5. Page 7, Figure 6: The quality of the figure should be improved; further, the font is too big in comparison with other figures and text.
  6. Page 8, Chapter 3.6 Cytotoxicity: What are the determined values (cell viability) for cytotoxicity biocompatibility according to ISO standard ?
  7. Page, 8, line 246: Check the English styling. I think that in the sentence “From the results, it can be said that the….” the comma should be deleted.
  8. Generally, the authors should try to slightly improve the discussion of their results (in some chapters) with analogous/similar studies; try to compare and contrast their findings. If it is possible to do this, it would improve the quality of the paper.

Reviewer 2 Report

1 Figure 1, MO images should be modified into “OM” images.

2 What is the effect of TPP on the biodegradation behavior? Why?

3 The discussion is needed to be improved, such as the TGA results and swelling tests.

4 In Fig. 6, the error bar should be added. How many times the tests are performed?

5 The reference should be strengthened. The following article suggested to be included in this manuscript in the biodegradation section: A core-shell structured ZIF-8@PDA-HA with controllable zinc ion release and superior bioactivity for improving poly-l-lactic acid scaffold, ACS Sus Eng, 2021.

Round 2

Reviewer 2 Report

It is ok now.